# Permeability of the Composite Magnetic Microcapsules Triggered by a Non-Heating Low-Frequency Magnetic Field

**DOI:** 10.3390/pharmaceutics14010065

**Published:** 2021-12-28

**Authors:** Ivan A. Burmistrov, Maxim M. Veselov, Alexander V. Mikheev, Tatiana N. Borodina, Tatiana V. Bukreeva, Michael A. Chuev, Sergey S. Starchikov, Igor S. Lyubutin, Vladimir V. Artemov, Dmitry N. Khmelenin, Natalia L. Klyachko, Daria B. Trushina

**Affiliations:** 1Shubnikov Institute of Crystallography of Federal Scientific Research Centre ‘‘Crystallography and Photonics” of Russian Academy of Sciences, 119333 Moscow, Russia; mikheev.av16@physics.msu.ru (A.V.M.); borodina@crys.ras.ru (T.N.B.); bukreeva_tv@nrcki.ru (T.V.B.); sergey.s.starchikov@gmail.com (S.S.S.); lyubutinig@mail.ru (I.S.L.); vladimir.artemov@gmail.com (V.V.A.); dirq@mail.ru (D.N.K.); trushina.d@mail.ru (D.B.T.); 2Department of Chemical Enzymology, Lomonosov Moscow State University, 119991 Moscow, Russia; veselov.mac@gmail.com (M.M.V.); klyachko@enzyme.chem.msu.ru (N.L.K.); 3Faculty of Physics, Lomonosov Moscow State University, 119991 Moscow, Russia; 4National Research Centre ‘‘Kurchatov Institute”, 123182 Moscow, Russia; 5Valiev Institute of Physics and Technology of RAS, 117218 Moscow, Russia; chuev@ftian.ru; 6Institute “Nanotechnology and Nanomaterials”, G.R. Derzhavin Tambov State University, 392000 Tambov, Russia; 7Department of Biomedical Engineering, I.M. Sechenov First Moscow State Medical University, 119991 Moscow, Russia

**Keywords:** polyelectrolyte microcapsules, triggered release, iron oxide nanoparticles, magnetic actuators, Brownian relaxation mechanism, non-heating low frequency magnetic field

## Abstract

Nanosystems for targeted delivery and remote-controlled release of therapeutic agents has become a top priority in pharmaceutical science and drug development in recent decades. Application of a low frequency magnetic field (LFMF) as an external stimulus opens up opportunities to trigger release of the encapsulated bioactive substances with high locality and penetration ability without heating of biological tissue in vivo. Therefore, the development of novel microencapsulated drug formulations sensitive to LFMF is of paramount importance. Here, we report the result of LFMF-triggered release of the fluorescently labeled dextran from polyelectrolyte microcapsules modified with magnetic iron oxide nanoparticles. Polyelectrolyte microcapsules were obtained by a method of sequential deposition of oppositely charged poly(allylamine hydrochloride) (PAH) and poly(sodium 4-styrenesulfonate) (PSS) on the surface of colloidal vaterite particles. The synthesized single domain maghemite nanoparticles integrated into the polymer multilayers serve as magneto-mechanical actuators. We report the first systematic study of the effect of magnetic field with different frequencies on the permeability of the microcapsules. The in situ measurements of the optical density curves upon the 100 mT LFMF treatment were carried out for a range of frequencies from 30 to 150 Hz. Such fields do not cause any considerable heating of the magnetic nanoparticles but promote their rotating-oscillating mechanical motion that produces mechanical forces and deformations of the adjacent materials. We observed the changes in release of the encapsulated TRITC-dextran molecules from the PAH/PSS microcapsules upon application of the 50 Hz alternating magnetic field. The obtained results open new horizons for the design of polymer systems for triggered drug release without dangerous heating and overheating of tissues.

## 1. Introduction

Nanotechnologies gain much attention due to the unique physical and chemical properties of its products having a ground-breaking impact in a range of fields such as chemistry, biology, and engineering. The fields of nanomedicine and drug delivery have undergone rapid development over the past decades. In contrast to the usual administration of a drug substance, targeted delivery of the encapsulated formulation allows to increase the selectivity of the therapeutic effect, ensure its prolonged action, and reduce side effects. To achieve this goal, tremendous efforts have been made to modifying the surface of carriers with antibodies and other ligands to increase their affinity to the target sites [1,2]. These specific interactions are proven to be effective in vitro [2,3], however, nonspecific uptake and clearance in vivo limit their performance [4]. An alternative approach is to use physical mechanisms [5,6,7], e.g., an external magnetic field to control the localization of the carriers and adjust their permeability.

Among a wide variety of the drug delivery vehicles, the polyelectrolyte multilayer capsules (PECs) obtained by the method of layer-by-layer adsorption of macromolecules on the surface of sacrificial colloid particles deserves special attention [8,9]. The PECs has been used for encapsulation of various biologically active substances [5,10] and delivery of anticancer agents [11,12], genetic material, antigens, and enzymes in vitro [13,14], as well as delivery of vaccines in vivo [15].

Development of magnetosensitive PECs has started since the 2000s [16,17], and in 2005 an enhanced accumulation of the magnetic nanoparticles (MNPs)-functionalized polymer capsules caused by magnetic field gradient was demonstrated for the first time in human breast cancer cells using a flow channel system modeling the bloodstream [18]. The latest works confirm that the external magnetic field is able to trap and navigate the magnetic PECs under physiological conditions of a living organism, whereas the capsules have no interference with the blood flow [19,20]. In addition to the remote navigation, MNPs are promising triggers for a controlled release of the encapsulated substances [9,21]. Hu et al. report on a dramatic increase in doxorubicin release caused by the rupture of the Fe_3_O_4_/poly(allylamine) capsules due to application of 50 kHz alternating magnetic field [22]. The work of Lui et al. has demonstrated another option: the amount of released doxorubicin from maghemite loaded sodium poly(styrene)/poly(allylamine) capsules increases by 10% at the same magnetic stimulus [23]. Variable permeability of the polymer shells to dextran molecules is described in [24], where capsules were irradiated by alternating electromagnetic field for 30 min at frequencies of 0.3–1 kHz.

The magnetic nanoparticles response to an alternating magnetic field primarily through two mechanisms is referred to as Neel and Brown relaxation [21,25]. Neel relaxation occurs due to the movement of the magnetic moment relative to the crystal lattice of the motionless MNPs and is attended with magnetic field energy dissipation and thus leads to particle heating. A vast majority of the published works is devoted to the study of the effect of a high frequency magnetic field on the PECs integrity, whereas its application has a number of disadvantages, e.g., difficulties in localization of the heating effect. Brown relaxation involves the movement of the whole nanoparticle relative to the surrounding medium-magneto-mechanical actuation [25,26]. In this case, the thermal effect of the field is extremely small and can be neglected. A particular relaxation mechanism dominates depending on the amplitude and frequency of the magnetic field, the particle size, and medium properties. Brown relaxation dominates when the single domain particles are exposed to the low frequency magnetic field (LFMF, <100 Hz). The power dissipation via Brown relaxation becomes predominant for the particles with a diameter greater than some critical value (e.g., the critical diameter for magnetite nanoparticles approximately is 14 nm [27]). The appeared mechanical vibration and motion of the MNPs can make an impact on the particle environment and induce mechanical stress, which causes the efficient release of the model macromolecules [25,28]. The magneto-mechanic disruption of a highly specific protein–protein complex and a protein–polymer complex during the exposure to a low-frequency magnetic field has been previously shown [29,30]. In addition, structural changes in the lipid liposomal membrane due to magneto-mechanical actuation have been demonstrated [31,32]. Thus, the magneto-mechanical actuation may hold great promise for affecting intermolecular bonds at the nanoscale, in particular, in the design of drug release systems. To date, the effect of LFMF on the permeability of magnetized polyelectrolyte capsules has hardly been studied, although they are prospective multipurpose containers. The first article on this topic states that the permeability of the capsules to the high molecular weight polymer and the low molecular weight doxycycline increase proportionally to the duration of LFMF exposure [33]. As it was shown in our previous study, a pulsed regime of the LFMF can also have a certain effect on the distribution of the labeled molecules in the capsule shell [34].

In this work, ferromagnetic citrate-stabilized iron oxide nanoparticles were synthetized and analyzed in terms of X-ray diffraction and Mössbauer spectroscopy. The crystallite size, calculated from the Scherrer equation, was found to be 15 nm which is in good agreement with the TEM observations. The composite microcapsules were tailored on the base of polyelectrolytes and the obtained nanoparticles. Investigation by SEM and TEM evidenced the uniform distribution of MNPs within the polymer capsule shell and the stability of the resulted capsules. Influence of LFMF on the capsule permeability was analyzed by measuring the optical density of the released TRITC-dextran during the LFMF exposure indicating successful application of the synthesized MNPs as local strain mediators due to the magneto mechanical effect.

## 2. Materials and Methods

### 2.1. Materials

Poly(allylamine hydrochloride) (PAH, 100 kDa), poly(sodium 4-styrenesulfonate) (PSS, 70 kDa), tetramethylrhodamine isothiocyanate–dextran (TRITC-dextran, 65–85 kDa), ethylenediaminetetraacetic acid disodium salt (EDTA), calcium chloride, sodium carbonate, ammonium thiocyanate, citric acid, sodium chloride (NaCl), iron(II) chloride tetrahydrate (FeCl_2_ 4H_2_O), and iron(III) chloride hydrate (FeCl_3_ H_2_O) were purchased from Sigma-Aldrich, St. Louis, MO, USA. Millipore Milli Q water (18.2 MΩ/cm) was used as an aqueous medium during all sets of experiments.

### 2.2. Magnetic Nanoparticles Synthesis

Magnetic nanoparticles were synthesized via chemical precipitation from di- and trivalent iron salt solutions under the base treatment according to R. Massart [35]. In detail, 2.7 g FeCl_3_*H_2_O and 1.0 g FeCl_2_*4 H_2_O were dissolved in 100 mL deionized water under heating at 80 °C with permanent stirring. Afterward, 6 mL of 0.25 M NH_4_OH was added to the mixture and the reaction was maintained at 80 °C for 30 min. The as-prepared nanoparticle suspension was stabilized with 10 mL of citric acid (30 mg/mL, pH 6.0) The precipitate was decanted using a magnet and washed with deionized water to remove the excess of citric acid three times.

### 2.3. Determination of the Iron Concentration

The concentration of iron ions was measured by colorimetric method. This method is based on the reaction between Fe^3+^ ions and with SCN^−^ followed by the formation of a ferric rhodanide complex.

For this aim, 0.1 mL of the sample was added to 0.4 mL of the mixture of HNO_3_ and HCl solutions (1:3). This mixture was kept for 1 h under sonication at 80 °C for magnetite dissolving and oxidation of Fe^2+^ ions to Fe^3+^. Then, 0.1 mL of 50% NH_4_SCN solution and 1.8 mL of deionized water were added to 0.1 mL of the analyzed sample, and the absorption spectra were measured. The absorption spectra of the standard solutions with the defined Fe^3+^ concentrations were measured, and a standard curve was plotted according to the absorbance data collected at 460 nm (Figure A1 in Appendix A).

### 2.4. Microcapsules Assembly

Microcapsules were prepared by the well-known Layer-by-Layer method [16] with slight modification. CaCO_3_ templates were prepared by pre-adding TRITC-dextran (0.4 mg/mL) to 0.33 M CaCl_2_ solution and then precipitated with 0.33 M Na_2_CO_3_. After washing with deionized water, PAH (2 mg/mL in 0.15 M NaCl) and PSS (2 mg/mL in 0.15 M NaCl) were subsequently assembled on the CaCO_3_ surface. After three layers formed by the polymers, the magnetic nanoparticles were used as a layer by the same procedure, and after this, another three polymer layers were adsorbed. At the last step CaCO_3_ cores were dissolved with 0.2 M EDTA solution by adding TRITC-dextran. This helped to decrease TRITC-dextran release from capsules due the diffusion balance during the core dissolution. After the core dissolution, microcapsules were washed three times with deionized water.

### 2.5. Methods

Samples were studied by scanning electron microscopy (SEM) using a JSM-7401F instrument (JEOL, Akishima, Japan). Secondary electron images were acquired using the lower secondary electron detector at an accelerating voltage of 1 kV and working distance of 8–9 mm without conductive coating. To conduct SEM measurements, water suspension of microcapsules was deposited on a silicon wafer and dried. Transmission electron microscopy (TEM) was performed using Tecnai Osiris microscope with accelerating voltage 200 kv and EDX element mapping provided by Super-X detector. Water suspension of samples was deposited on lacey carbon coated copper TEM grids and dried. TEM images were obtained by conventional electron microscopy and by scanning-transmission electron microscopy (STEM) as well. ImageJ version 1.52p was used for size distribution analysis of nanoparticles.

The ζ potential and size distribution were determined by dynamic light scattering (DLS) measurements using a Zetasizer Nano ZS automatic analyzer (Malvern Panalytical Ltd., Malvern, UK) at 25 °C. Sodium chloride solution (0.15 M) with the adjusted pH was used to prepare the dispersions of nanoparticles for the zeta potential measurements. Prior to the hyrdodynamic size analysis, nanoparticles were dispersed in an ultrapure water and sonicated for 5 min at room temperature.

Fluorescence of TRITC–dextran was visualized using a Leica TCS SP laser scanning confocal microscope (Leica Microsystems, Wetzlar, Germany) equipped with a 100× immersion objective with a digital aperture of 1.4.

X-ray diffraction pattern of the nanoparticle powder was measured using Rigaku Miniflex 600 with CuKα radiation (λ = 1.5418 Å). Sample was scanned at a Bragg 2Θ angle between 20 and 80 at room temperature (step size 0.02, time per step 1 s).

Mössbauer spectra of magnetic nanoparticles were obtained in transmission geometry using an MS-1104Em spectrometer. The radiation source was ^57^Co(Rh). Isomer shifts were calibrated against a standard absorbent 30-μm thick α-Fe foil. Low-temperature measurements were carried out in a modernized version of the liquid-free closed-cycle helium cryostat [36].

The magnetization curve was obtained at room temperature using a LakeShore 7407 vibrating magnetometer. The applied magnetic field varied from 5 kOe to −5 kOe.

The iron ion concentration was analyzed by spectrophotometry (Perkin Elmer, Lambda 650, UV/VIS Spectrometer, Waltham, MA, USA).

The in situ release of TRITC-dextran from the microcapsules was evaluated by measuring the optical density during the influence of magnetic field. For this, magnetic field generator Astra-M-SP (Nanodiagnostika LLC, Tambov, Russia) with built-in light source AvaLight-DHc (Avantes, Apeldoorn, The Netherland) and spectrometer AvaSpec 2048 (Avantes, Apeldoorn, The Netherland) was used. The scheme of the set-up is presented in Figure A2. The AC generator modulates harmonic current of specified frequency and voltage. This signal gets through to electromagnet and as a result sinusoidal magnetic field forms. The light beam from the light source passes through the thermostatic cuvette located inside the electromagnet air gap and the value of optical density is measured by the spectrometer. The samples were placed in the magnetic field generator in a glass cuvette with a total volume of each sample of 0.4 mL. Evolution of optical density was measured at the wavelength 520 nm corresponding to TRITC-dextran absorption maximum (Figure A3). The irradiation time for each measurement was approximately 30 min, which was optimal for reaching the constant values of the optical density curves. First 3–4 points were measured without LFMF, then switching on LFMF, and all the other points were measured under the irradiation.

## 3. Results

### 3.1. Magnetic Nanoparticles Characterization by TEM, DLS and Zeta-Potential Analysis

The shape and size of iron oxide MNPs were studied by TEM (Figure 1a). TEM showed that the shape of the nanoparticle is close to spherical with an average diameter of *d*_0_ = 12.25 nm and a standard deviation of Δ*d* = 3.37 nm (Figure 1b). According to TEM data, the relative width of the particle size distribution is Δ*d/d*_0_ = 0.27. From the point of view of magnetic properties, particles of this size (*d* < 20 nm) are considered single-domain [37]. DLS measurements (Figure 1c) indicated the MNPs with unimodal distribution and the average hydrodynamic radius of 39 ± 21 nm (PDI 0.24). The size difference is well-known to be explained by the fact that DLS measures the hydrodynamic diameter of the NPs including core plus any molecule attached or adsorbed on its surface. TEM provides information of the sample in the dry state, while DLS requires measurements in the solvated state, where solvent molecules are associated with the particles. Moreover, TEM responds to the electron-density of the particles not taking into account any protecting or stabilizing layer that may surround the particles. In our case, the MNPs are surrounded by a layer of the citrate ions, which is clearly reflected in the movement of the particles in the suspension effecting diffusion coefficient, which is related to hydrodynamic size via the Stokes–Einstein equation.

We studied the influence of pH on Zeta-potential values of the MNPs (Figure 1d). The results show that as the pH approaches neutral value, the nanoparticles acquire the maximum surface charge, which practically does not change when the medium is transferred to an alkaline pH. This is explained due to an increase of citric ions deprotonation with a transition to the more alkaline region as pKa of citric acid is 6.4 [37]. Zeta-potential at pH 7.0 is –21 ± 1 mV, indicating the more charged surface of magnetic nanoparticles as a result of citric acid deprotonation. The subsequent deposition of nanoparticles on the polymer layers was carried out from a suspension with a neutral pH.

### 3.2. X-ray Diffraction

All reflections on the XRD pattern of the nanoparticles can be indexed into a cubic phase with a spinel-type structure space group *Fd*-3*m* (Figure 2). Observed reflections correspond to magnetite Fe_3_O_4_ or/and maghemite γ-Fe_2_O_3_ phases with similar structures. However, in the case of small nanoparticles it is hard to distinguish only these two compounds from XRD. The iron valence state in these phases is different and Mössbauer spectroscopy as a sensitive method of the iron valence state can distinguish these phases more accurately. The estimation of the crystallite size by Scherrer equation gives the value of 15 nm which agrees well with the TEM observations.

### 3.3. Mössbauer Spectroscopy

In the crystal structures of magnetite Fe_3_O_4_ and maghemite γ-Fe_2_O_3_, iron ions occupy octahedral [B] and tetrahedral (A) sites. The cationic distribution in magnetite Fe_3_O_4_ is (Fe^3+^)_A_ [Fe^3+^Fe^2+^]_B_ O_4_. At room temperature, at the [B] sites, a fast electronic exchange occurs between Fe^3+^ and Fe^2+^, which leads to the averaged valence of Fe^2.5+^ at [B] sites. At T = 120 K, the electron exchange at the [B] sites is “frozen”, and the Verwey transition takes place. In maghemite, all iron ions are in the Fe^3+^ state, and it can be considered as non-stoichiometric magnetite, where some of the octahedral cationic sites are vacant (Fe^3+^)_A_ [Fe_5/3_^3+^
_1/3_]_B_ O_4_. The ratios of Fe ions in (A) and [B] sublattices in γ-Fe_2_O_3_ and Fe_3_O_4_ are 1:1.67 and 1:2, respectively.

The Mössbauer spectra of the obtained nanoparticles are shown in Figure 3. At room temperature, the spectrum exhibits hyperfine magnetic splitting of lines, which indicates a magnetically ordered state of iron atoms in the sample. Note that the line in the spectrum is strongly skewed toward the inner part of the spectra. This shape of Mössbauer lines is determined by the spectrum of thermal excitations of the magnetic moments of nanoparticles and indicates a slow relaxation of the moments in comparison with the lifetime of the nucleus in the excited state [38].

The analysis of the spectra was carried out within the model of the magnetic dynamics of ferromagnetic nanoparticles [38]. An example of the use and detail description of such a model is presented elsewhere [39]. This approach allows one to describe not only qualitatively but also quantitatively the shape of the spectra over the entire temperature range. In addition to the standard parameters of hyperfine interaction, such as isomer shift *δ*, hyperfine magnetic field at the iron nucleus *H_hf_*, and quadrupole coupling constant *q*, our computational model also allows us to obtain the average magnetic anisotropy energy *KV* for the given ensemble of nanoparticles and the relative width of the size distribution of magnetic domains Δ*D/D*_0_. Here *K* is the constant of magnetic anisotropy, and *V* is the average volume of the magnetic domain.

When approximating the Mössbauer spectra at different temperatures, two magnetic components corresponding to iron ions in the octahedral [B] and tetrahedral (A) magnetic sublattices were used. The ratio of the areas of the components corresponding to the A and B sites was fixed either as 1:1.67 or 1:2. The model was fitted for all spectra of the sample simultaneously. This allows one to find the most self-consistent solution for each spectrum. The model demonstrated good agreement with the experiment (Figure 3) when using the ratio 1: 1.67. This indicates that the synthesized nanoparticles have the structure of γ-Fe_2_O_3_ maghemite. The parameters obtained are presented in Table 1.

At a temperature of 7 K, the obtained parameters of the isomer shift *δ_A_* = 0.376 (2) mm/s and *δ_B_* = 0.471 (3) mm/s correspond to Fe^3+^ ions in the high-spin state (3*d*^5^, *S* = 5/2) at tetra- and octa-sites of γ-Fe_2_O_3_, respectively. No traces of ferrous ions related to magnetite were observed in Mössbauer spectra. The maghemite γ-Fe_2_O_3_ phase was also observed in previous work, where similar polyelectrolyte microcapsules modified by iron oxide nanoparticles were studied [41]. It should be noted that maghemite formation could be due to the magnetite oxidation during the synthesis. Table 1 also contains parameters for bulk maghemite to compare with nano-compounds.

Note the difference between the values of the hyperfine magnetic field *H_hf_* in [B]-sites and the magnetic anisotropy constant in bulk maghemite and in the studied nanoparticles (Table 1). The magnetic anisotropy constant we calculated is in good agreement with the literature data, which are (1–2.5) × 10^5^ erg/cm^3^ for γ-Fe_2_O_3_ nanoparticles about 10 nm in size [42]. The parameter of the relative size distribution width of magnetic domains correlates well with the results of magnetometry, shown below, and transmission electron microscopy.

### 3.4. Magnetometry

The magnetization curve for the nanoparticles is shown in Figure 4. One would expect that iron oxide nanoparticles with a size of about *d*_0_ = 12.5 nm should have superparamagnetic properties, which are crucial for application in biomedicine. Indeed, the general form of the magnetization curve of our sample is typical for a superparamagnet, in which there is no hysteresis. However, we found that in the region of low magnetic fields the magnetization curve on an enlarged scale shows a slight hysteresis with a remanence of about 2.2 emu/g and a coercive force of 38 Oe. This may be due to the particle size distribution. In this case, small particles (most of them) are in a superparamagnetic state, while larger particles can be in a spin-blocking or magnetically ordered state, and they are responsible for the observed hysteresis.

It is known that superparamagnetic blocking temperature depends on the characteristic time of the experiment. The time-scale of Mössbauer spectroscopy is much less than the time-scale of magnetometry measurement. As shown above, in the Mössbauer spectrum at room temperature, we do not observe doublet lines characteristic of the superparamagnetic state. Mössbauer spectroscopy indicates that the spin-blocking temperature of nanoparticles is above room temperature.

The theoretical processing of the magnetization curve was carried out within the generalized Stoner–Wohlfarth relaxation model [43,44,45,46]. It was assumed that each magnetic particle is uniformly magnetized with magnetization *M*_0_, and the magnetic anisotropy has an axial form and its energy is described by the parameter *KV*, where *K* is the constant of magnetic anisotropy, and *V* is the average volume of the magnetic particle.

In the model, the particles have the shape of elongated ellipsoids with the anisotropy axis coinciding with the axis of rotation of the ellipsoid. We assume a random distribution of the directions of the anisotropy axes of particles in the sample. Accordingly, for each particle, the anisotropy axis is randomly directed in space.

In the initial state (with uniaxial anisotropy), the magnetic moment of the particle is in one of two possible positions in the local energy minima. If the external magnetic field (*H_ex_*) is less than the critical field of the particle magnetization reversal (*H_C_*), then the magnetic moments of the particles can jump under the influence of thermal excitations.

Depending on the direction of *H_ex_*, the magnitude and rate of change of the field, the anisotropy energy, and the direction of the easy axis of the particle magnetization, these jumps will have a certain characteristic frequency *p*_0_. When *H_ex_* > *H_C_*, only one local energy minimum remains, which determines the direction of the particle magnetic moments. Our model takes into account the size distribution of particles, which is determined by the Gaussian function.

Analysis of the magnetization curves of nanoparticles within this model allows one to determine such parameters as the saturation magnetization *M*_0_, the magnetic anisotropy constant *K*, the critical magnetic field of the particle magnetization reversal *H_C_* = 2*K*/*M*_0_, the average magnetic particle volume *V*, the magnetic anisotropy energy *KV*, the average particle diameter *D*_0_, the relative width particle size distribution Δ*D/D*_0_, the relaxation frequency parameter *p*_0_, and the parameter responsible for the interaction between particles *b* [44]. The generalized Stoner–Wohlfarth model takes into account the interaction of particles in the mean-field approximation:(1)$$H\left(t\right)=\frac{\Delta H}{\Delta t}t+b\;M\left(t\right)$$
where $\frac{\Delta H}{\Delta t}$ is the rate of change in the external magnetic field during the experiment, *M* is the instantaneous magnetization of the particle, *b* is the mean field parameter responsible for the interaction of nanoparticles. A detailed formalism of the model describing all the calculated parameters is presented in [43,44,45,46] and is beyond the scope of this article. Despite its simplicity, the model makes it possible to qualitatively and quantitatively describe the experimental magnetization curves without resorting to excessive parameterization of the problem, for example, introducing a more complex type of anisotropy [43,44,45,46]. The numerical values of the parameters obtained as a result of processing the experimental magnetization curve by the proposed model are presented in Table 2.

It was found that the average size of the magnetic domain is in good agreement with the average size of nanoparticles, which was obtained from the TEM data. This suggests that the resulting nanoparticles are predominantly single-domain. The positive value of the parameter *b* indicates the ferromagnetic nature of the interaction between the particles. Not a high value of this parameter (*b* = 0.087) indicates that this interaction is rather weak [43,44].

Accordingly, to the theoretical concepts, the synthesized MNPs can implement Brownian relaxation of the magnetic moment, and can be used as local strain mediators due to the magneto-mechanical effect.

### 3.5. Fabrication and Characterization of Composite Microcapsules

The microcapsules were synthesized by sequential adsorption of oppositely charged polymers and MNPs on the surface of the colloidal CaCO_3_ particles. The reversal of Zeta-potential during the layers deposition indicates the successful formation of multilayers shell (Figure A4). At the same time, adsorption of the MNPs as the 4th layer practically has no influence on the colloidal stability of the system. To recharge any remaining positive sites on the surface after nanoparticles deposition, we adsorb polyanion as a 5th layer. As could be seen from Figure A4 the absolute value of zeta-potential increased after PSS coating. The microcapsules with the following shell composition were synthetized: PAH/PSS/PAH/MNP/PSS/PAH/PSS. Figure 5 shows TEM image of the single PAH/PSS/PAH/MNP/PSS/PAH/PSS microcapsule with an element mapping. The presence of carbon, oxygen, sulfur, and sodium over the entire area of the capsule indicates a uniform distribution of the polyelectrolytes during the shell formation. The signals from iron (clearly associated with the MNPs) exhibit the uniform distribution of the nanoparticles over the microcapsule shell without aggregates. The energy dispersive X-ray (EDX) microanalysis shows the following elemental abundances (Table 3); however, this data should be treated with caution due to possible carbon contamination and the presence of a carbon sublayer of the TEM grid in the analyzed area.

Typical SEM-images of the individual capsules and magnified capsule surface with observable MNPs are presented on Figure 6a,b. The confocal fluorescent image of the PAH/PSS/PAH/MNP/PSS/PAH/PSS capsules loaded with TRITC-dextran (Figure A5) confirms its successful entrapment within the capsules.

The capsules with average diameter of 3–5 µm can be injected into the blood flow drug targeting to the kidney without any harming effect, and the blood flow remains intact [19,47]. Despite this fact, we cannot consider microcapsules to be optimal, their size must be reduced to nanoscale before use in vivo. As the miniaturization technique has already been developed [48], below we provide a proof of concept that the magneto-mechanical actuation of nanoparticles in the capsule shell can facilitate the release of cargo molecules.

Concentration of the magnetic nanoparticles in the shell was calculated from concentration of iron ions (see Materials and Methods Section 2.3). This parameter is very important to control the behavior of the composite capsules under the magnetic field irradiation. Concentration of Fe^3+^ ions in the sample of microcapsules loaded with TRITC-dextran and in the control sample (without TRITC-dextran) is 2.6 ± 0.5 and 2.8 ± 0.4 µg/mL, respectively.

### 3.6. Influence of LFMF on Release of TRITC-Dextran

It is known, that molecules with high molecular weight cannot penetrate the shell of polyelectrolyte microcapsules [49,50]. In this work we use TRITC-dextran with molecular weight 65–85 kDa as a model substance to monitor changes in release induced by LFMF. We suggest that TRITC-dextran could penetrate the polyelectrolyte shell when composite microcapsules are exposed to the LFMF magnetic field due to the magneto mechanical actuation of the MNPs. Figure 7a demonstrates the changes in optical density due to release of TRITC-dextran from the capsules under the LFMF exposure. As one can observe, the optical density decreases in time for all studied LFMF frequencies from 30 to 150 Hz, which is associated with the gradual precipitation of the composite capsules to the electromagnet poles. The curve corresponding to the exposure to 50 Hz LFMF differs from the whole bunch of curves indicating the presence of another process besides precipitation. All measurements were performed on the wavelength of TRITC-dextran absorption maximum, which indicates its release from the microcapsules. Comparison of the control non-loaded sample and the capsules with TRITC-dextran indicated a credible difference in the optical density curves (Figure 7b), which confirms the successful application of the synthesized MNPs to serve as local strain mediators due to the magneto mechanical effect. The absorbance curves corresponding to the influence of other frequencies are presented in Figure A6.

Optical density curves can be described by the function with three components (2):(2)$$OD\left(t\right)=S_{0}+A*\left(1-e^{-v_{1}t}\right)+B*e^{-v_{2}t},$$
where *S*_0_ is a background level due to the scattering and absorption of molecules and particles in the sample, which do not attract to poles of electromagnet, constant *A* is a final level of released TRITC-dextran in sample, $v_{1}$—velocity of establishment of the equilibrium concentration of TRITC-dextran in cuvette volume, constant *B* corresponds to the light scattering on microcapsules, $v_{2}$—velocity of attraction of microcapsules to magnet poles. So, the second term in the Equation (2) is associated with the release of TRITC-dextran, and third term describes the attraction of microcapsules to the walls of the cuvette adjacent to the electromagnet. In case of capsules without TRITC-dextran the component associated with release is not applicable.

The dynamic part of optical density is described by the sum of two components, which compete with each other. The attraction to the walls of the cuvette reduces the concentration of microcapsules in the area of beam path, which reduces the scattering of light by the microcapsules. LFMF, forming in the electromagnet air gap, in practical is not uniform. The inhomogeneity of the generated magnetic field, according to the manufacturer, is no more than 5%. Intensity of magnetic field increases near the core that leads to attraction of MNPs. Due to the oscillation of the magnetic field, the emerging forces alternately attract the capsules to the poles of the magnet. However, due to the inertial motion of the capsules and the rotation of magnetic nanoparticles in the shell, the oscillation of the magnetic field cannot fully compensate for the initial displacement of the microcapsules. In contrast, the release of TRITC-dextran from microcapsules increases the optical density through absorption. The process of release from microcapsules and diffusion of TRITC-dextran into the beam region are slower than the attraction of microcapsules to the walls of the cuvette. Thus, at the beginning of the process, the optical density curve decreases. Then the role of the release process begins to grow, and the second and third components become balanced.

To calculate the parameters from the Equation (2), namely *S*_0_, *A*, *B*, $v_{1}$, $v_{2}$, curves for the samples without TRITC-dextran have been fitted first. The calculated parameters from this fitting (*S*_0_, *B*, $v_{2}$) have been used in fitting curves for the samples loaded with TRITC-dextran. Parameters S_0_, *B* are common for all samples and do not depend on frequency of magnetic field. The calculated values of those parameters are *S*_0_ = 0.585 ± 0.002, *B* = 0.831 ± 0.005. Others calculated parameters are shown in Table 4. The goodness of fit was evaluated by adjusted r-squared parameter. Calculated values for used samples presented in Table A1.

The value of parameter A is of a special interest. As can be seen from the Table 4, the value of A for the sample after irradiation in LFMF with a frequency of 50 Hz is two or more times higher compared to other frequencies. The anomalously high values of the $v_{1}$ and errors for the samples irradiated with the magnetic field at 112 and 149.5 Hz are associated with the very small value of the parameter A. Values of parameter $v_{2}$ are close for all samples. This indicates that value of magnetic field frequency influence only the TRITC-dextran release from the microcapsules and has no influence on the other process in cuvette under irradiation by magnetic field.

## 4. Conclusions

We demonstrate that permeability of composite magnetic microcapsules can be triggered by non-heating low frequency magnetic field. For this purpose, the single-domain γ-Fe_2_O_3_ maghemite magnetic nanoparticles were synthetized by precipitation method by R. Massart. The characteristics of the tailor-made nanoparticles were analyzed in order to obtain an effective drug delivery system with remote release of the encapsulated component. The results demonstrate the perspectives of the nanoparticles application as the magneto-mechanical actuators due to their ability to the Brownian relaxation. The multilayer composite microcapsules were developed on the base of sacrificial vaterite templates from PAH and PSS polyelectrolytes and the synthesized nanoparticles, which are uniformly distributed over the shells. The TRITC-dextran-loaded microcapsules permeability triggered by LFMF was investigated at 100 mT and frequencies of 30–150 Hz range. A low-frequency magnetic field of 50 Hz was found to be the most pronounced to selectively increase the permeability of magnetically sensitive microcapsules to the high molecular weight TRITC-dextran. We present here a proof-of-concept and expect this effect to be much stronger for low molecular weight substances, which are mainly used in medicine. Our findings could provide a perspective application of the microcapsules with LFMF-triggered permeability for controlled release of drugs without dangerous heating and overheating of the biological tissues.

## Figures and Tables

**Figure 1 pharmaceutics-14-00065-f001:**
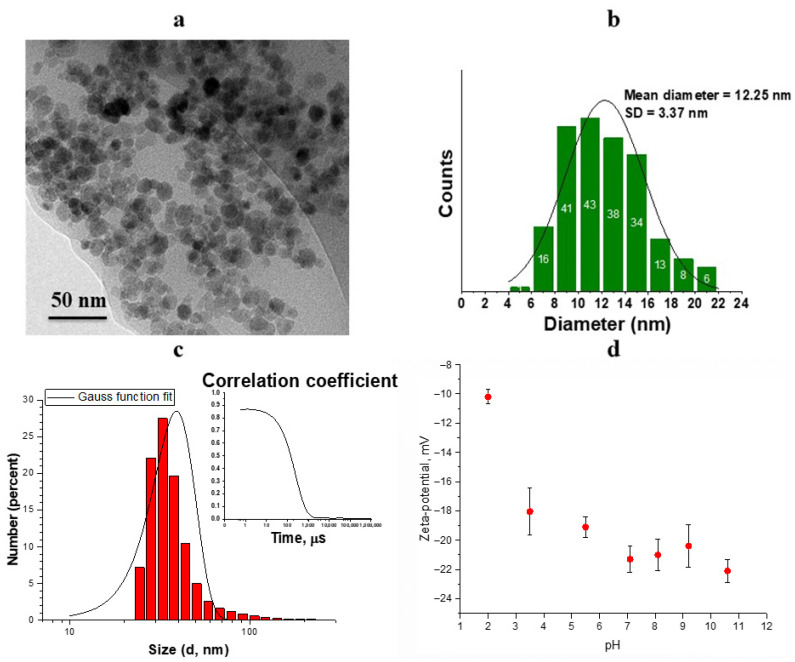
(**a**) TEM image of the magnetic nanoparticles, (**b**) size distribution of nanoparticles from TEM data approximated by Gaussian function, (**c**) DLS data of the MNPs hydrodynamic size distribution, (**d**) Zeta-potential of MNPs as a function of pH.

**Figure 2 pharmaceutics-14-00065-f002:**
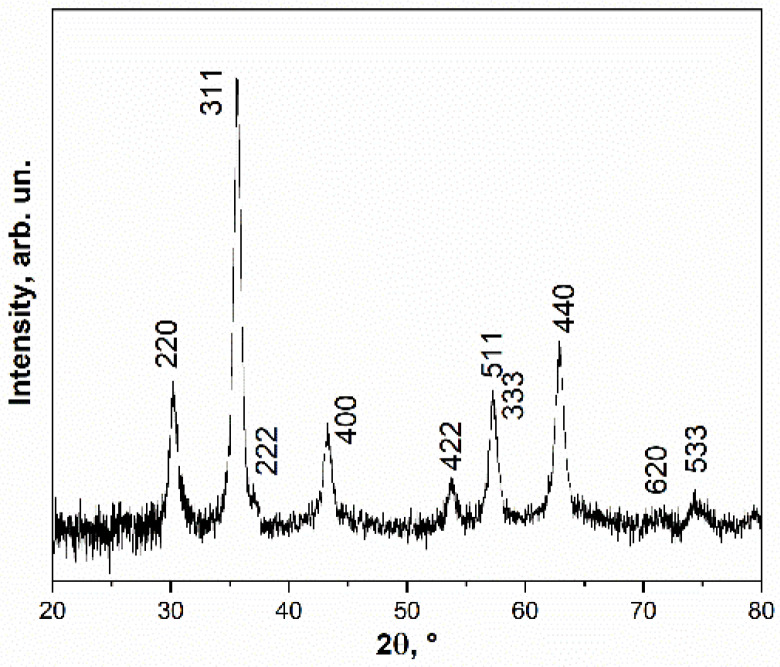
XRD pattern of nanoparticles.

**Figure 3 pharmaceutics-14-00065-f003:**
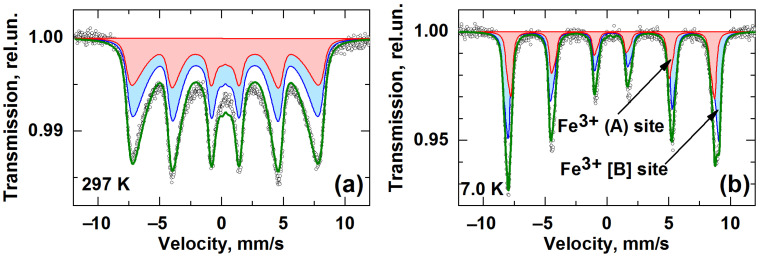
Mössbauer spectra of iron oxide nanoparticles at 297 K (**a**) and 7.0 K (**b**). Points are experimental values; solid lines are calculated spectra.

**Figure 4 pharmaceutics-14-00065-f004:**
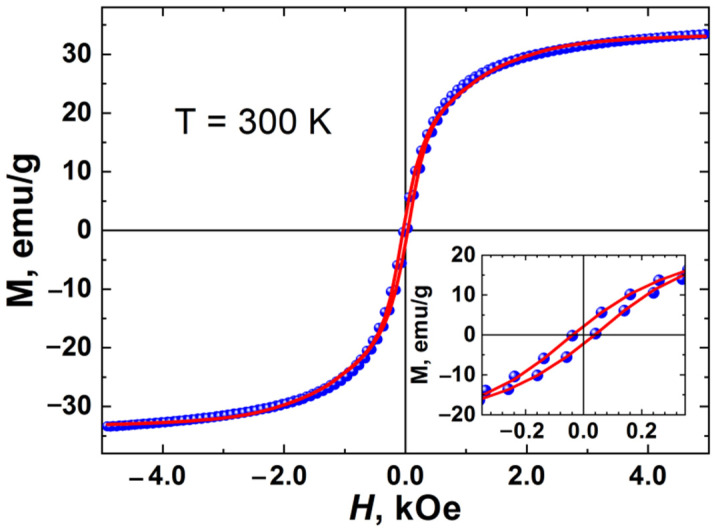
Experimental magnetization curve of a nanoparticles at room temperature (blue circles). The solid (red) lines show the curves calculated within the generalized Stoner–Wohlfarth model. The inset shows part of the curve at low magnetic fields.

**Figure 5 pharmaceutics-14-00065-f005:**
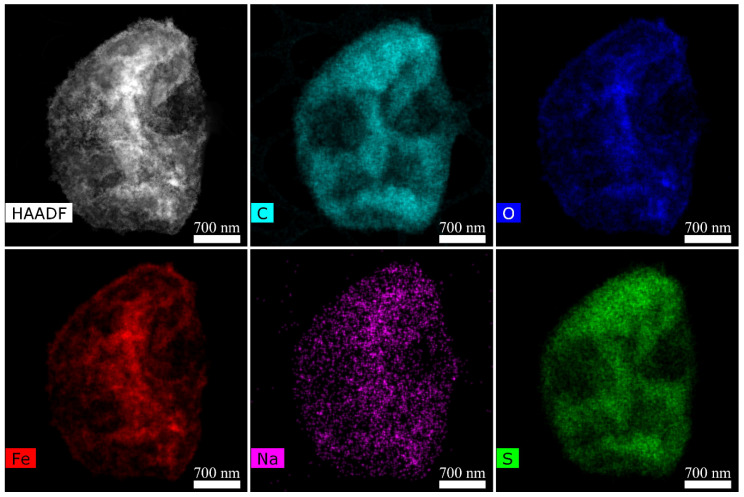
STEM image and elemental mapping of the PAH/PSS/PAH/MNP/PSS/PAH/PSS microcapsule.

**Figure 6 pharmaceutics-14-00065-f006:**
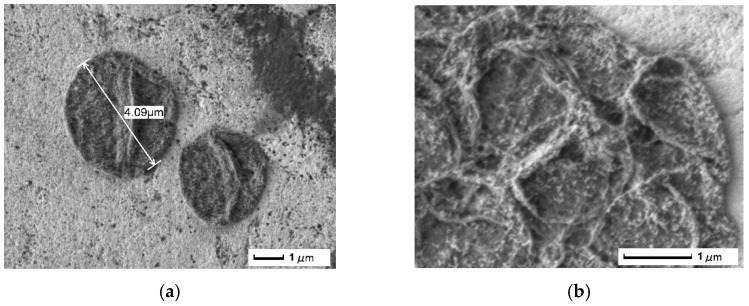
SEM images of separate PAH/PSS/PAH/MNP/PSS/PAH/PSS microcapsules (**a**) and of a magnified microcapsule shell with detectable MNPs (**b**).

**Figure 7 pharmaceutics-14-00065-f007:**
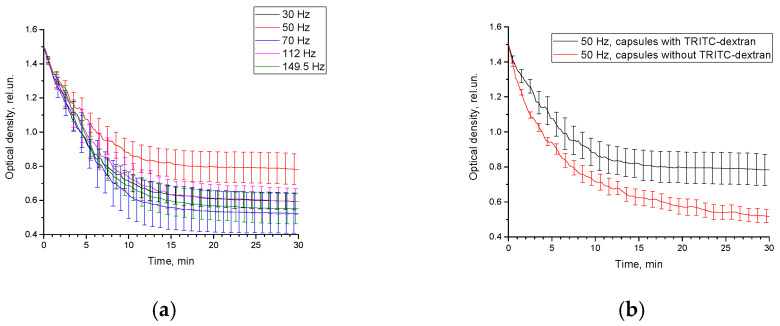
Optical density curves for suspension of capsules loaded with TRITC-dextran under exposure of 100 mT alternating magnetic field with frequencies 30, 50, 70, 120, 150 Hz (**a**), optical density curves for suspension of capsules loaded with TRITC-dextran and for control sample under exposure of 100 mT, 50 Hz LFMF (**b**).

**Table 1 pharmaceutics-14-00065-t001:** Results of the analysis of the Mössbauer spectra of nanoparticles. *δ_A,B_* is the isomer shift for iron atoms in the (A) and [B] lattice sites; *q* is the quadrupole coupling constant; *H_hf_* is the hyperfine magnetic field at the ^57^Fe nucleus.

	*δ_A_*, mm/s	*H_hf A_*, T	*δ_B_*, mm/s	*H_hf B_*, T	*q*, mm/s	*KV*/*k_B_*, K	*K*, erg/cm^3^	ΔD*/D*_0_
Bulk γ-Fe_2_O_3_ at 8 K [40]	0.37 (2)	51.4 (2)	0.47 (2)	52.8 (2)	-	-	* 4.7 × 10^4^	-
Nano (12 nm) at 7 K	0.37 6(2)	51.95 (5)	0.471 (3)	54.30 (5)	−0.35 (2)	1064 (73)	2.0 × 10^5^	0.268 (20)
Nano (12 nm) at 297 K	0.307 (3)	49.54 (20)	0.322 (20)	49.91 (20)	−0.35 (5)	484 (20)	0.9 × 10^5^	0.276 (5)

* *K* value for bulk γ-Fe_2_O_3_ was taken from [40].

**Table 2 pharmaceutics-14-00065-t002:** Results of the analysis of the magnetization curve.

*M*_0_, emu/g	*KV*/*k_B_*, K	*K*, erg/cm^3^	*H_c_*, kOe	*D*_0_, nm	Δ*D*/*D_0_*	*p*_0_, s^−1^	*b*	*d*_0_, nm	Δ*d*/*d_0_*
33.44 (2)	1125 (15)	2.08 (1)·10^5^	2.42 (2)	11.26 (7)	0.26 (1)	1510 (5)	0.087 (3)	12.25	0.27

**Table 3 pharmaceutics-14-00065-t003:** Results of the EDX microanalysis.

C	O	S	Fe	Na
50.0 (4.5)	13.2 (1.3)	6.9 (0.7)	29.7 (2.8)	0.1 (0.08)

**Table 4 pharmaceutics-14-00065-t004:** Calculated parameters for function (2).

Frequency, Hz	A, rel.un.	$v_{1},\;s^{-1}$	$v_{2},\;s^{-1}$
30	0.097 ± 0.001	0.221 ± 0.012	0.191 ± 0.003
50	0.184 ± 0.003	0.159 ± 0.010	0.160 ± 0.002
70	0.085 ± 0.008	0.111 ± 0.028	0.175 ± 0.002
112	0.014 ± 0.004	0.140 ± 0.136	0.150 ± 0.002
149.5	0.078 ± 0.005	0.200 ± 0.051	0.137 ± 0.002

## Data Availability

Not applicable.

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
