# Peer review of "Permeability of the Composite Magnetic Microcapsules Triggered by a Non-Heating Low-Frequency Magnetic Field"

_pharmaceutics, 2021, doi:10.3390/pharmaceutics14010065_

Round 1

Reviewer 1 Report

The authors present their study on polyelectrolyte microcapsules and the influence of alternating magnetic field on the release of encapsulated molecules, triggered by the presence of magnetic nanoparticles in the structure of the microcapsules. The subject is very interesting and many of the characterizations reported in the manuscript are accurate and help to depict the structure and properties of the microcapsules. However, I have a number of comments to bring to the attention of the authors.

  1. There is some confusion in the organization of the paragraphs of the paper. Sometimes there is redundancy. For example, sect. 2.5 could be included in 2.6, since the same method is described also there. Also, in 3.5, which should contain only the results, the synthesis of microcapsules, already discussed in Sec. 2, is mentioned again.
  2. Some techniques are mentioned but not defined. For example, the use of the acronym DLS should be accompanied by the mention of the dynamic light scattering technique. Also, some additional information on the use and purpose of the Z-potential analysis could be useful for the reader.
  3. The analysis of the magnetometry results (Sec. 3.4) deserves some clarification. First of all, the authors comment Fig. 4 and conclude that the sample is ferromagnetic, since it has a small hysteresis. I find that the main feature of the magnetization curve is typical of superparamagnetism. This property, typical of magnetic nanoparticles, is never cited in the paper.
    Also, in the next analysis, using the generalized Stoner-Wohlfarth model, the authors hypothesize the existence of domains, initially not identified a priori. Therefore, the magnetic system is considered as a whole, without taking in consideration that it is constituted by particles. Only at the end, as a conclusion, the domains are identified with the particles. It is said that there is uniaxial anisotropy (without giving a justification for that) and each domain can only have one of the two opposite orientations. This is a limiting assumption, in the real case, since the system is made of nanoparticles and therefore, we expect that for each of them the anisotropy axis can be randomly oriented. The author should also provide some more detail about the model used, otherwise the values reported in Table 2 seem a little arbitrary. The real meaning of many such parameters cannot be understood without indicating the basic relations they enter.
  4. I found Sec. 3.6 a little confusing. The author claim that absorbance has 3 contributions (Eq. 1). The first is a constant value (parameter So) due to “stable” molecules. The second is a function, raising with time (parameters A and n1), due to the release ot TRITC. The last is a decreasing (with time) component due to “attraction of microcapsules to magnet poles” (parameters B and n2). First of all, it is not clear what these magnet poles are. Fig SI 3 doesn’t show the presence of any magnet poles. The magnetic field is generated by a coil, not an electromagnet. Besides, the field is alternated in time and it should be pretty uniform in space. There should be no drift of the molecules due to the magnetic field. The authors should better describe this phenomenon. It would be interesting to measure also the time dependence of the signal without the alternated field. This would really separate the first constant parameter So and possible contributions due to sedimentation with respect to the contribution due to the field. Moreover, In table 4, no unit is indicated for the various quantities (beside frequency). Also, It is not clear why So, B and n2 are different for the various frequencies. The authors claim they are similar but they are different instead, if we look the error values reported in the table. They should have been calculated from the reference sample (with no TRITC) once for all. In addition, n1 is not very different from n2, which, associated with the fact that A is small compared to B, does not really allow to identify the contribution of release compared to that of precipitation. In fact, the overall trend of the absorption curves is always decreasing with time and an increase due to the release of TRITC triggered by the a.c. field is hardly evident. Finally, the only dependence on frequency should be reasonably ascribed only to the parameter n1, not even A, if the samples used in the different experiments at different frequency are identical. In fact, A should be determined by the amount of TRITC initially loaded in the microcapsules, and n1 by the rate of release that can be conditioned by the field frequency. The authors also state in lines 411-412 that n1 is determined by A. This is a nonsense, since they are independent parameters in the fitting. The claimed dependence on the parameters n2 and B is incomprehensible too (line 413).
    If the reported results were real and not an artifact, i.e. if the equilibrium concentration of released TRITC depends on frequency, this could be confirmed by the following experiment. First, a “non-optimal” frequency is used and the absorption is observed to decrease in time for 30 minutes following the claimed mechanism described by the parameter n2 until the asymptotic value is reached. Then, the frequency is switched to 50 Hz so to verify if an increased released amount of tracer (and optical absorption) is observed. Did the authors try this?
  5. Finally, the English style should be checked. A number of sentences do not sound correct. Some of them seem to have a missing verb. See for example lines 104-106, 132-134, 135, 136, 150-151, 161-163, 376-378, 392-393, 423-424, 429-431. Also, “measure” is misspelled in lines 183 and 189. The first word in line 226 should be an adverb, not an adjective.

For all the above comments, I believe that the authors should revise the manuscript before it can be accepted for publication in Pharmaceutics.

Reviewer 2 Report

As the studied systems have already been published in ref 34 by the same research group, the main novelty and point of interest  is the study of macromolecule release (Labelled Dextran) induced by a low frequency magnetic field. My main doubt in what concerns drug delivery applications is the big size of the proposed particles 4microns and this limitation was not mentioned or discussed in the manuscript. There are also some figures that are either equal or have the same data as figures in ref 34. I also have some doubts on using the same value of S0 parameter in equation ** for samples with and without TRITC-Dextran as it should be higher for the labelled particles due to TRITC absorption. Please consider an annotated pdf file of the manuscript with all my comments, corrections or suggestions (made with okular software running on kde/linux).

Reviewer 3 Report

The manuscript "Permeability of the composite magnetic microcapsules triggered by a non-heating low-frequency magnetic field" describes the obtaining of some microcapsules with iron oxides, which can release the carried drug by applying an external magnetic field.

English should be polished and typos corrected. Across the manuscript put the numbers as proper indices in formulas e.g. Fe3O4 (row 67).  Please use L and mL instead of l and ml as symbols.

In introduction part, some important and new reports about magnetic nanoparticles should be added to show clear background (e.g. rows 40-51): doi: 10.3390/ma14071612, 10.3390/molecules26082189 and 10.3390/pharmaceutics13091356

Authors can improve the quality of the manuscript by DSC measurements.

Did authors monitored the microcapsules sample temperature during experiment?

Is there an estimation of loaded TRITC-dextran quantity (and entrapment efficiency)?

What can authors say about the stability of the system in time?

The manuscript is well-organized, devoid of more serious errors and may be interesting for readers. Therefore, I suggest publishing the manuscript “Permeability of the composite magnetic microcapsules triggered by a non-heating low-frequency magnetic field” after major correction in the journal Pharmaceutics.

Reviewer 4 Report

Reviewing of the paper “Permeability of the composite magnetic microcapsules triggered by a non-heating low-frequency magnetic field

This manuscript describes the synthesis of polyelectrolyte microcapsules integrated with single domain maghemite nanoparticles and describes the effects of the magnetic field with different frequencies on the permeability of the microcapsules and the consequent release of TRITC-dextran.

The originality of the work and the scientific relevance can be considered on the average, in consideration of the fact that the use of magnetic nanoparticles coupled with a magnetic field to obtain a release of an active compound by micro-nanoparticles is well known and described in literature.

The manuscript appears to be well organized, with good acknowledgement of the work of others in the references even if some more experimental details and new experiments must be added to better justify the results here presented.

I think this paper can be reconsidered for publication only after major revisions.

Please add the following changes to the manuscript:

Page 4 from line 217, sub-chapter “2.6. Methods”: the description of the experiments herein reported is too poor; for example there is no mention of the buffers used in DLS experiments and ζ potential determination. Likewise there is no mention on how samples have been prepared for SEM, TEM and X-ray diffraction experiments.My suggestion is to expand the chapter content adding much more information concerning all the aspects involved in these experiments.

Page 4 from line 192, sub-chapter “3.1. Magnetic nanoparticles characterization by TEM, DLS and zeta-potential analysis”: “The shape and size of iron oxide MNPs were studied by TEM (Fig.1a). TEM showed that the shape of the nanoparticle is close to spherical with an average diameter of d0 = 12.25 nm and a standard deviation of Δd = 3.37 nm (Fig. 1b). According to TEM data, the relative width of the particle size distribution is Δd/d0 = 0.27. DLS measurements (Fig.1c) indicated the MNPs with unimodal distribution and the average hydrodynamic radius of 196 77±20 nm. The size difference is well-known to be explained by the fact that DLS measures the hydrodynamic diameter of the NPs including core plus any molecule attached or adsorbed on its surface…”.
Looking at the values correlated with the data herein reported, even considering the standard deviation, what it seems to me is that the difference between TEM and DLS data is too high (a ratio of more than 16 between the two!!!). This difference in measured size can not be ascribable to the hydration shell or to whatever interaction whit small molecules present in the solution. In my opinion the enormous size given by DLS in comparison to TEM is due much more to aggregation processes, probably favored by the unknown buffer used for these analysis.
My suggestion is to repeat the DLS measurements using different buffers and after extensive sonication of the samples and at the same time add another analytical technique to evaluate the dimension of the magnetic nanoparticles in solution, for example using Nanoparticle Tracking Analysis (NTA) technique (Malvern Panalytical NanoSight NS300).

Page 5 Figure 1d: looking at the graph present in Figure 1d I find unusual the authors can't reach a zero and positive value using magnetic nanoparticles. Probably this result is strongly dependent on the buffers used during the experiments. My suggestion is to use 10 mM NaClO4 titrated at different pH as a buffer, to ensure the proper electrophoretic mobility to samples in ζ potential determination.

Extensive editing of English language and style are required, because I found many typing and grammar errors during the text. I strongly suggest to revise the entire document from this point of view.
Only some of the errors I found are reported here below as examples:

  • page 2 line 54: “…on the surface of sacrificial colloid particles deserves special attention…”;
  • page 3 line 106: "…which in good agreement with TEM observations. ";
  • page 3 line 127: “Then 6 ml NH4OH (0.25 M) was added by a syringe.”;
  • page 3 line 128: “…10 ml of citric acid (pH 6.0) was 128 added.”;
  • page 3 line 135: “For this aim, 0,1 ml of the sample mixed with 0.4 ml mixture of HNO3…”;
  • page 3 line 136: “This mixture placed in ultrasonic bath at 80°C for 1 hour for oxidation …”;

Round 2

Reviewer 1 Report

In the response to my comments on sec. 3.4, the authors say: "By definition, a material is superparamagnetic if its magnetization curve has no hysteresis". This is not the definition of superparamagnetism. Even a bulk ferromagnetic material could, in principle, have zero hysteresis. That would be a perfect soft ferromagnet. In that case, the complete random reorientation of domains at zero field would account for the phenomenon. On the contrary, superparamagnetism occurs in small particles when the thermal energy overwhelms the anisotropy energy. The authors, although admitting that at the given average size the particles should be superparamagnetic, insist in their statement and affirm that they are not. They give no explanation to support this anomaly. The entire approach of sect. 3.4 is based on this. As shown in Fig. 4, there are only small remanence and coercivity which are observed only in the enlargement of the inset. Incidentally, those values are less than half the values reported in the authors’ reply.  The authors assume that all particles behave in the same identical way and conclude that if there is hysteresis they are not superparamagnetic. On the contrary, it should be clear that, due to particle size distribution (or partial clustering of few particles), a very small portion of the specimen could be responsible for the observed hysteresis, whereas the largest part is superparamagnetic.

In line 242, the authors added the sentence stating that the particles are single domain. Therefore, as I tried to suggest in my previous comment, in the discussion regarding the Stoner Wohlfarth model, they should avoid using the term “domains” and replace it with the term “particles”. The rest of my comment was not aimed to suggest the existence of “multiaxial” anisotropy, as interpreted in the authors’ reply. Instead, I was concerned on the fact that, if the sample is considered as a whole, and it is constituted by domains with two possible orientation of the domain magnetization, the reader is led to believe that there is uniaxial anisotropy of the entire sample (same easy axis for all particles). In the long reply, the authors explain (eq. 5 in the reply) that the particles axes are randomly oriented (it makes sense if particles are considered, due to their shape, but not if a whole system constituted by domains is considered instead), but in the text has never been reported. I believe a sentence should be added to clarify that the particle axis is randomly oriented.

The reply to my comment of Sec. 3.6 is still not clear. The authors affirm: ”The electromagnet is connected to the AC generator, which supplies a current through the coils”. It seems that there is an electromagnet AND the coils. Also, later in the response, the authors say: ”The inhomogeneity of the sinusoidal magnetic field formed in the electromagnet core gap is no more than 5%”. In the figure I see only a coil; no electromagnet gap. Therefore, I still find confusing the use of the term “magnet poles” (instead of “ends of the coil”) if the magnetic field is generated by a coil. I would rather explain that segregation is induced by the magnetic field gradient which is significant close to the ends of the coil.

Later, the authors emended their fitting, leading to the values reported in table 4. If a common value for S0 and B is assumed now, then this should be stated in the text. The corresponding values for S0 and B should be reported in the text and it is not necessary to include them in the table.

Reviewer 2 Report

The paper can be accepted after some minor corrections as the authors have complied to most of my comments/concerns:

- Figure 1 as a lot of repetitions with different images but the same analysis results, although with slightly different plots. I think just the first set is needed

- in Figure 5 new images were included as requested in my review, but the old ones are still there. The old images should be removed as the repetitions from ref. [34] are still on the manuscript.

- In the sequence of the image in line 430-431 I feel that a comment should be included regarding the micrometer size of the nanocapsules. The authors gave a reasonable explanation in the response letter but did not include in the manuscript:

" The capsules with average diameter of 3-5 µm can be injected into the blood flow drug targeting to the kidney without any harming effect, and the blood flow remains intact (10.1021/acsami.6b15811, 10.1016/j.jconrel.2020.11.051). Despite this fact, we cannot consider microcapsules to be optimal, their size must be reduced to nanoscale before use in vivo. Here we present a proof of concept that the magneto-mechanical actuation of nanoparticles in the capsule shell can facilitate the release of cargo molecules. As we have already done the size minimization (10.1016/j.colsurfb.2018.06.033), the next step will be testing the nano-sized magneto sensitive capsules in the experiments in cell cultures in vitro and in animals in vivo." 

Author Response

  • - Figure 1 as a lot of repetitions with different images but the same analysis results, although with slightly different plots. I think just the first set is needed

We removed repetitions from the text

  • in Figure 5 new images were included as requested in my review, but the old ones are still there. The old images should be removed as the repetitions from ref. [34] are still on the manuscript.

We removed old image from the text

  • - In the sequence of the image in line 430-431 I feel that a comment should be included regarding the micrometer size of the nanocapsules. The authors gave a reasonable explanation in the response letter but did not include in the manuscript:

" The capsules with average diameter of 3-5 µm can be injected into the blood flow drug targeting to the kidney without any harming effect, and the blood flow remains intact (10.1021/acsami.6b15811, 10.1016/j.jconrel.2020.11.051). Despite this fact, we cannot consider microcapsules to be optimal, their size must be reduced to nanoscale before use in vivo. Here we present a proof of concept that the magneto-mechanical actuation of nanoparticles in the capsule shell can facilitate the release of cargo molecules. As we have already done the size minimization (10.1016/j.colsurfb.2018.06.033), the next step will be testing the nano-sized magneto sensitive capsules in the experiments in cell cultures in vitro and in animals in vivo." 

Thanks to the referee for comment. We added this in text

Reviewer 3 Report

The authors have responded to my comments and have addressed all my concerns therefore, I suggest publishing the paper in the current form.

Author Response

Thanks for reviewer for comments

Reviewer 4 Report

The authors have suitably addressed my concerns.

Author Response

Thanks for reviewer for comments

Round 3

Reviewer 1 Report

I still have one fundamental objection:

There is still a debate if the particles are, or are not, superparamagnetic. Again, the composition and size of the magnetic particles should indicate that they are. The authors, in their last version, in section 3.4, introduce the problem and only partially consider my previous comments. Their conclusion (lines 315-317 of the last version) is still that the particles are in the blocked state, although admitting that the remanence is very small. Their statement relies on Mossbauer experiments which shows a typical ferromagnetic response. Unfortunately, the authors seem not to be aware that the blocking temperature depends on the characteristic time of the experiment. In the fundamental Neel's theory dated 1949, and all the many works treating superparagnetism, it is emphasized the role of the Neel's relaxation time, which is a function of anisotropy constant, volume, and temperature. In order to determine if the particles are in the superparamagnetic regime or in the blocked regime, the time of the experiment must be compared with the Neel's relaxation time. In other words, it must be determined if during the characteristic time of the experiment, the magnetization has enough time to flip between the two directions allowed by the uniaxial anisotropy. It turns out that the characteristic time of the Mossbauer technique is less than 10-7 seconds (related to the time of the nuclear transition), whereas the characteristic time of a conventional magnetometry experiment is several seconds (typically taken as 102 seconds). The dramatic difference of time scale simply implies that the magnetization curves are typical of a superparamagnet (excluding a small fraction: otherwise the remanence would be very much larger) even if during the Mossbauer experiment the particles behaves as blocked since the magnetization, only in the latter case, does not have enough time to "fluctuate" during the nuclear transition. There is no need to assume that the particles are ferromagnetic since in the generalized Stoner-Wohlfarth model the effect of temperature is taken into account: as stated in lines 333-334 "... the magnetic moments of the particles can jump under the influence of thermal excitation". The average time of these jumps is exactly the Neel's relaxation time I was mentioning before.
